# Characterization of Histone Modifications in Late-Stage Rotator Cuff Tendinopathy

**DOI:** 10.3390/genes14020496

**Published:** 2023-02-15

**Authors:** Kayleigh J. A. Orchard, Moeed Akbar, Lindsay A. N. Crowe, John Cole, Neal L. Millar, Stuart M. Raleigh

**Affiliations:** 1Centre for Sports, Exercise and Life Sciences, Coventry University, Coventry CV1 5FB, UK; 2School of Infection and Immunity, College of Medicine, Veterinary and Life Sciences, University of Glasgow, Glasgow G12 8TA, UK

**Keywords:** rotator cuff tendinopathy, epigenetics, histone modifications, chromatin immunoprecipitation, extracellular matrix, inflammation

## Abstract

The development and progression of rotator cuff tendinopathy (RCT) is multifactorial and likely to manifest through a combination of extrinsic, intrinsic, and environmental factors, including genetics and epigenetics. However, the role of epigenetics in RCT, including the role of histone modification, is not well established. Using chromatin immunoprecipitation sequencing, differences in the trimethylation status of H3K4 and H3K27 histones in late-stage RCT compared to control were investigated in this study. For H3K4, 24 genomic loci were found to be significantly more trimethylated in RCT compared to control (*p* < 0.05), implicating genes such as *DKK2*, *JAG2*, and *SMOC2* in RCT. For H3K27, 31 loci were shown to be more trimethylated (*p* < 0.05) in RCT compared to control, inferring a role for *EPHA3*, *ROCK1*, and *DEFβ115*. Furthermore, 14 loci were significantly less trimethylated (*p* < 0.05) in control compared to RCT, implicating *EFNA5*, *GDF6*, and *GDF7*. Finally, the TGFβ signaling, axon guidance, and regulation of focal adhesion assembly pathways were found to be enriched in RCT. These findings suggest that the development and progression of RCT is, at least in part, under epigenetic control, highlighting the influence of histone modifications in this disorder and paving the way to further understand the role of epigenome in RCT.

## 1. Introduction

Rotator cuff (RC) disease is a prevalent musculoskeletal pathology affecting both athletes and nonathletes and between 30–50% of the population over the age of 50 [1]. RCT is considered a progressive disorder that begins as acute tendinitis and progresses to chronic RCT with degenerative conditions, and so this term is used to describe a combination of pain and impaired performance that is associated with injury to the tendons of the RC [2,3]. The diagnosis of RCT is difficult to reach. This, in part, is due to the poor association between structural changes that are identified through imaging and clinical symptoms, so the definitive cause of this disease remains uncertain [4]. There is increasing evidence that RCT is a multifactorial disorder that can manifest through a combination of extrinsic, intrinsic, and environmental factors, which contribute to its development and progression, including genetics and epigenetics. 

Research in related conditions, such as Achilles tendinopathy (AT), has shown associations between RCT and single-nucleotide polymorphisms (SNPs) in a variety of genes, including defensin-β1 (*DEFβ1*), estrogen-related receptor β (*ESRRβ*), fibroblast growth factors and receptors (*FGF3, FGF10*, and *FGFR1*) [5,6]. Additionally, several genes involved in soft tissue homeostasis have also been associated with RCT, including *COL1A1*, *COL1A2*, *COL3A1*, *COL5A1*, and *COL5A2* [7]. Variation within the *MMP* and *TIMP* families of genes may also contribute to RC tears [8,9]. 

Whilst the role of genetics in RCT is recognized, the epigenetic profile of RCT has yet to be fully elucidated, and at present there is no study that investigates the role of histone modifications in RCT. Histone modifications can influence the structure of chromatin, which has key roles in the positive and negative regulation of gene expression. For example, heterochromatin is associated with gene repression as well as high levels of methylation on histone H3 and lysine 27 (H3K27) [10], while euchromatin is associated with gene activation and poised promoters as well as trimethylation of H3K4 [10,11]. Several studies have investigated the role of epigenetics in RCT, identifying significant DNA methylation differences that influence the expression of *MMP1*, *MMP9*, *MMP13*, and *TIMP2* as well as hypomethylation of *HOXC4*, *HOXC8*, and *HOXA3* in RC satellite cells, which may influence their adipogenic potential [12]. The TGFβ signaling pathway has also been implicated in RCT, and *TGFβ1* induces the expression of type I collagen through the epigenetic inhibition of *DNMT1*, *DNMT3a*, and global DNA activity, which results in the demethylation of the *COL1A1* promoter [13]. Furthermore, multiple miRNAs have been associated with RCT, including *miR-26a-3p*, *miR-22-3p*, *miR-7a-5p*, *let-7g-5p*, and *miR-26b-5p*, which were predicted to have regulatory control over genes that have been identified as dysregulated in tendinopathy, such as, *COL1A1*, *COL1A2*, and *COL3A1* [14]. In tendinopathic tissue, *miR-23a* and *-26* expression is decreased, and these miRNAs have been shown to regulate interleukin-6 and other proinflammatory cytokines, which may allow increased inflammatory mediator production that could negatively affect the integrity of tendons [14,15]. 

However, the specific interactions and functional roles of miRNAs, and epigenetic mechanisms in general, in RCT is still unclear. Chromatin immunoprecipitation (ChIP) sequencing (ChIP-Seq) allows genome-wide discovery of protein–DNA interactions, such as histone modifications [16], and can therefore be used to investigate the underlying mechanisms of gene regulation and reconstruct the epigenetic signature of key genomic regions in various diseases, including RCT [16,17]. In the present study, we sought to characterize the trimethylation status of H3K4 and H3K27 in late-stage RCT compared to control. These histone modifications are associated with euchromatin (H3K4) and heterochromatin (H3K27) and have been associated with a wide range of biological processes. An over-representation analysis was also conducted on the genes of interest obtained from this study to identify whether biological functions or processes are over-represented (enriched) in RCT. We hypothesized that genes involved in the maintenance of the extracellular matrix and inflammatory and immune responses would be implicated in this study. 

## 2. Materials and Methods

### 2.1. Participants and Human-Derived Cell Culture

Late-stage RC (supraspinatus) tendon samples and control hamstring (semitendinosus) tendon samples were provided by the School of Infection and Immunity at the University of Glasgow. Informed consent was obtained from participants, and the use of tissue for research was approved through the West of Scotland Research Ethics Service (REC 16/WS/0207). A total of 5 supraspinatus tendon samples were collected from patients undergoing arthroscopic shoulder surgery for RC tears and were harvested from within 1.5 cm of the edge of the tear using the standard 3-portal technique as described by Wu et al. [18]. The mean age of patients was 50 (*n* = 5, with ages ranging from 28 to 68). In addition, 4 hamstring tendon (semitendinosus) samples were collected from patients undergoing routine arthroscopic anterior cruciate ligament reconstruction surgery, and the mean age of patients was 26 (*n* = 4, with ages ranging from 21 to 37). Human-derived tendon cells were explanted from both hamstring tendon tissue and supraspinatus tendon tissue, and cultures were provided by the University of Glasgow and maintained in RPMI 1640 (ThermoFisher; Cheshire, UK) supplemented with 10% fetal calf serum, 100 U/mL penicillin, and 100 µg/mL streptomycin at 37 °C in a humidified atmosphere of 5% CO_2_ until confluent. Once confluent, cells were washed with warmed, sterile dPBS, and 1 mL of Accutase (ThermoFisher; Cheshire, UK) was used to dissociate the cells for 5 min at 37 °C. All flasks that corresponded to the same donor were then condensed into a 15 mL falcon tube, and dPBS was added to result in a final volume of 10 mL. Samples were then centrifuged at 400× *g* for 5 min, the supernatant was discarded, and the cell pellet was resuspended in 1 mL of ice-cold PBS. Methanol-free formaldehyde was then added at a volume that equates to a final concentration of 1%. The samples were gently rotated for 10 min at room temperature (RT) until 100 µL glycine was added to quench the cross-linking reaction of formaldehyde and then further rotated for 5 min at RT. After incubation, samples were centrifuged at 600× *g* for 5 min at 4 °C, the supernatant was discarded, and the cell pellet was resuspended in lysis buffer (Appendix A) (in a volume suitable for the number of cells present, i.e., 100 µL for 10^6^ cells/mL) and incubated on ice for 10 min. 

### 2.2. Sonication

Cell lysates were sonicated using a Diagenode Biorupter Pico (Diagenode; Seraing, Belgium,). Lysates were subjected to 15 cycles of sonication (30 s on, 30 s off) at a temperature of 4 °C to produce chromatin fragments of 200–600 base pairs (bp). To determine the size of the chromatin fragments, decrosslinking solution (Appendix A) was produced, and lysates were incubated at 65 °C for 1 h. Remaining chromatin sample volumes were then stored at −80 °C until required for immunoprecipitation. Chromatin fragments were then purified using the MinElute PCR Purification Kit (Qiagen; Manchester, UK) and following the protocol provided by the manufacturer. Chromatin fragments were then eluted in a 10 µL elution buffer (Appendix A), and the chromatin fragment size was determined using a 1.5% agarose gel. 

### 2.3. Immunoprecipitation

Preparation of beads for immunoprecipitation (Invitrogen Dynabeads Protein A for Immunoprecipitation; Paisley, UK) was as detailed in Appendix A. 

Chromatin samples were taken from storage at −80 °C, quickly thawed at 37 °C for 2 min, and then placed directly onto ice. This was followed by 85 µL per sample being transferred to 1.5 mL LoBind Eppendorfs (Sigma Aldrich; Gillingham, UK) and topped up to a final volume of 350 µL with a dilution buffer without SDS (Appendix A). Chromatin samples were then centrifuged at 13,000 rpm for 10 min, and the supernatants were transferred to clean 1.5 mL LoBind Eppendorfs. Additionally, 35 µL of each chromatin sample was taken for use as the input sample, which was stored at −20 °C until required for further processing. Antibody-coated beads were then resuspended with 50 µL of dilution buffer without SDS (Appendix A) per sample, which was then added to 300 µL of chromatin and subsequently rotated overnight at 4 °C. After incubation, samples were placed onto a magnetic rack and the dilution buffer was removed. Samples were then resuspended in 500 µL of wash buffer I (Appendix A) at 4 °C and placed into an Eppendorf rotator for 5 min, followed by the magnetic rack for 1 min. The supernatant was then discarded. Samples were then resuspended in 500 µL of wash buffer II (Appendix A) at 4 °C, and the above process was repeated twice. This was followed by samples being resuspended in 500 µL of wash buffer III (Appendix A) at 4 °C, and the above process was repeated twice. Samples were then resuspended in 100 µL of elution buffer (Appendix A) and 2 µL of Proteinase K (20 mg/mL stock; Invitrogen; Paisley, UK). Input samples were then removed from storage at −20 °C and topped up to 100 µL with 63 µL of elution buffer (Appendix A) and 2 µL of Proteinase K. 

Samples were decrosslinked using a thermocycler preset to 1 h at 55 °C, followed by 65 °C overnight, and then held at 4 °C. Samples were then placed into a magnetic rack for 1 min. The supernatant containing the chromatin fragments was placed into a 1.5 mL Eppendorf, and the beads were discarded. Chromatin samples were purified using the MinElute PCR Purification Kit (Qiagen; Manchester, UK) following the protocol provided by the manufacturer, with minor changes. Chromatin samples were eluted in 25 µL of the elution buffer (from the MinElute PCR Purification Kit; Qiagen; Manchester, UK) instead of 10 µL. The quality of purified samples was assessed using the Qubit 4 Fluorometer (Qiagen; Manchester, UK), and samples were stored at −20 °C until Illumina library preparation.

### 2.4. Library Preparation and Sequencing

The NEBNext Ultra II DNA Library Preparation for Illumina Kit (NEB; Hitchin, UK) was used to prepare the samples for sequencing following the protocol provided by the manufacturer for minor changes. Dual Index Primers (NEB; Hitchin, UK) were used in the NEXNext Ultra II DNA Library Preparation for Illumina protocols (NEB; Hitchin, UK), and primer combinations and sequences used were as detailed in Appendix A. Samples were then sent to the University of Glasgow Polyomics Facility (Glasgow Polyomics, University of Glasgow, Wolfson Wohl Cancer Research Centre, Garscube Campus, Bearsden, G61 1BD), where they were sequenced using an Illumina NextSeq500 Instrument (Illumina; SY-415-1002). 

### 2.5. Bioinformatic Analysis

Initial quality control was performed using FastQC version 0.11.8 [19]. A Galaxy server [20] was used to carry out further analysis, and files were trimmed using Trimmomatic (Galaxy version 0.36.5) [21]. Trimmed sequences were then mapped to the Homo sapiens genome (GRCh38) by Bowtie 2 (Galaxy version 2.3.4.2) [22,23], and the output files were concerted from SAM files to smaller binary files (BAM) and filtered to remove any low-quality, nonunique, or unmapped reads using SAMTools (Galaxy version 1.8) [24]. Peak finding was achieved using the Model-Based Analysis of ChIP-Seq (MACS)-2 (Galaxy version 2.1.1.20160309) [25,26]. Called peak files were converted to BED format, counted, and merged using BEDTools (version 2.27.1) [27] on VM Ubuntu (version 20.04.2). Coverage was performed by Mr J. Cole using BEDTools (version 2.21.1) [27], and RStudio [28] was then used to run DESeq2 (version 1.30.0) [29]. Using DESeq2 (version 1.30.0) [29] and RStudio [28], the ChIP-Seq bound signals were normalized by a median of ratios method. This produced a table of normalized counts, which was used for initial analysis of the data using ggplot2 (version 3.3.3) [30], reshape2 (version 1.4.4) [31], and amap (version 0.8.18) [32]. Normalized peak counts were classified as significant using a criterion of adjusted *p*-value of less than 0.05 and a log2fold of more than 1, producing a table of significant peak counts. Homer (version 4.11) [33] was then used on VM Ubuntu (version 20.04.2) to annotate peaks to the Homo sapiens genome (GRCh38). GOEnrich was used to analyze pathways of identified genes, and this was achieved using RStudio [28], BiocManager (version 1.30.10) [34], ClusterProfiler (version 3.18.0) [35], and Pathview (version 1.30.1) [36]. Finally, DAVID (version 6.8) [37,38] was used for pathway analysis of the Entrez IDs of the genes that were identified for H3K4me3 and H3K27me3. 

## 3. Results

### 3.1. Trimethylation Status of H3K4 in Late-Stage RCT

Through normalization with DESeq2, 98,359 peak observations were identified for *H3K4me3*. Using a predetermined criterion of adjusted *p*-value of less than 0.05 and a log2fold value of more than 1, 25 of these peak observations were flagged as significant. Based on this, and as seen in Figure 1A, a volcano plot was produced to visually identify significant peaks. In addition to this, and as seen in Figure 1B, a heatmap was produced to gain further insight into the significant peak observations across samples. This revealed that the presence of trimethyl H3K4 significantly varied between sample conditions and that more trimethyl H3K4 was significantly present in disease compared to control. Peak annotation was achieved using Homer, and the 25 significant peaks were associated with 25 genes that are located nearby, as seen in Appendix A. Out of these 25, 23 were identified as protein-coding and 2 were identified as noncoding RNAs (ncRNA). Based on this, pathway and gene set enrichment analysis was conducted using GOEnrich and DAVID to look at the enrichment of the genes that were identified. However, no pathways were identified through either of these methods. 

### 3.2. Trimethylation Status of H3K27 in Late-Stage RCT

Through normalization with DESeq2, 81,107 peak observations were identified for H3K27me3. Using a predetermined criterion of adjusted *p*-value of less than 0.05 and a log2fold value of more than 1, 145 of these peak observations were flagged as significant. Based on this, and as seen in Figure 2A, a volcano plot was produced to visually identify significant peaks. In addition to this, and as seen in Figure 2B, a heatmap was produced to gain further insight into the significant peak observations across samples. This revealed that the presence of trimethyl H3K27 significantly varied between sample conditions and that more trimethyl of H3K27 was significantly present in disease compared to control. Peak annotation was achieved using Homer, and the 145 significant peaks were associated with 45 genes that are located nearby, as seen in Appendix A. Out of these 45, 29 were identified as protein-coding, 10 were identified as pseudogenes, and 6 were identified as ncRNA. Based on this, pathway and gene set enrichment analysis using GOEnrich and DAVID was conducted to look at the enrichment of the significant genes that were identified. From GOEnrich, and as seen in Figure 3A, a bar plot was produced to look at enriched terms as well as a gene-concept network, as seen in Figure 3B, identifying 10 pathways, with two having adjusted *p*-value of 0.011. As detailed in Table 1, DAVID functional annotation analysis also identified 10 pathways that were enriched, and from this, the TGFβ pathway, activin receptor pathway, axon guidance, regulation of focal adhesion assembly, positive correlation of pathway-restricted SMAD protein phosphorylation, and SMAD protein signal transduction were identified, which correlated with the results from GOEnrich. 

## 4. Discussion

This study identified, for the first time, that the trimethylation status of both H3K4 and H3K27 histones differed significantly between rotator cuff tissue specimens from individuals with RCT compared to control tendon tissue, suggesting an epigenetic component to RCT.

For H3K4, 24 genomic loci were found to have significantly more trimethylation in RCT compared to control, and one genomic locus was significantly less trimethylated in RCT compared to control. Using peak annotation, these loci were associated with 25 genes that are located in, or nearby, the loci of interest. A number of the genes that were identified have been associated with the regulation of skeletal muscle, cell proliferation, and migration as well as other musculoskeletal disorders. For example, *JAG2* was identified as being proximal to a significant locus that had increased trimethylation of H3K4 in RCT. *JAG2* encodes the Notch Ligand Jagged2, which has a role in the Notch pathway and is a central regulator of skeletal muscle during adult myogenesis. The Notch pathway is involved in the activation of satellite cells, which have important roles in the fusing with, and forming of, muscle fibers after skeletal muscle injury through proliferation, migration, and differentiation [39]. The absence of Notch in muscle satellite cells has been shown to lead to depletion of these cells due to premature differentiation [40]. Finally, homozygous and compound heterozygous mutations in *JAG2* are identified families that have autosomal recessive limb-girdle muscular dystrophy-27, which suggests a disease mechanism that is related to Notch pathway dysfunction [39]. However, the role of this gene in RCT is currently unclear, and further research is required. *SMOC2* was also identified to be located near a significant locus that had increased trimethylation of H3K4 in disease, and this gene encodes a member of the SPARC family that is highly expressed during wound healing. This gene promotes matrix assembly and can stimulate the proliferation of endothelial cells, angiogenesis, and migration [41]. Interestingly, using single-cell transcriptomics, De Micheli et al. identified three tenocyte groups that expressed unique extracellular matrix (ECM)-binding genes, including *SMOC2*, in mouse Achilles tendon [42]. Unfortunately, no studies at present have investigated the role of *SMOC2* and RCT, so further research is required.

For H3K27, 31 genomic loci had significantly more trimethylation in RCT compared to control, and 14 loci were found to have significantly less trimethylation in control compared to RCT. Using peak annotation, these loci were associated with 45 genes that are located in, or nearby, the loci of interest. A number of the genes that were identified have been associated with ECM organization, including cell adhesion, migration, and cytoskeletal organization, as well as the formation of focal adhesions, tendon cell differentiation, inflammation, and wound healing. For example, *EPHA3* and *ROCK1* were both identified to be located near, or in, a significant locus that had increased trimethylation of H3K27 in disease, while *EFNA5* was identified to be proximal to a locus that had decreased trimethylation of H3K27 in disease. Both *EPHA3* and *EFNA5* have roles in cell–cell adhesion, cytoskeleton organization, and migration [43], while *ROCK1* has roles in actin cytoskeleton organization and remodeling of the ECM, including cell adhesion, motility, proliferation, and apoptosis [44]. These genes were enriched in the regulation of focal adhesion assembly pathway, which refers to the formation of focal adhesions (FAs) that provide strong adhesion to the matrix, allowing mechanical tension to be transmitted from cells, across the plasma membrane, and to the external environment [45,46]. This can influence differentiation or cell survival by altering gene expression as well as regulation of pathways necessary for cell migration, growth, proliferation, wound healing, and tissue repair [46,47]. FAs are also stabilized by actin–myosin contractility, which enhances adhesion strength and generates cellular traction, thus leading to cell spreading and migration [48]. Interestingly, the loss of *ROCK1* was found to result in failure of immature adhesion complexes to form mature stable FAs in human keratinocytes, which resulted in decreased wound closure. This is likely a consequence of the failure of FAs to mature, which deprives the cell of actin–myosin contractile forces required for movement [49]. If the loss of *ROCK1* has a similar effect in RCT, it could be speculated that the delayed tissue repair and wound healing associated with RCT is a consequence of a lack of mature FAs.

*DEFβ115* was also identified to be located in, or near, a locus that had increased trimethylation of H3K27 in RCT. Human β-defensins are small cationic peptides that have a role in managing the microbial interface by contributing to epithelial barriers, primarily in epithelial surfaces, as well as activating the immune system by modulating signaling pathways and inflammatory responses [50,51]. Genetic association studies have previously identified associations between RCT and SNPs in *DEFβ11*, which has a role in preventing the colonization of epithelial surfaces by microbes [6,52].

Finally, *GDF6* and *GDF7* were proximal to loci that have decreased trimethylation of H3K27 in RCT. GDFs can induce the differentiation of tendon cells through the SMAD signaling pathway, and a reduction in the presence of these factors can result in a reduction in collagen fiber size and delayed healing in response to injury [53]. A study by Wong et al. identified a potential role of *GDF6* in human patellar tendon fibroblasts in remodeling of the ECM in adult tendon [54], and both *GDF6* and *GDF7* have been shown to induce the formation of new connective tissue as well as enhance tendon repair [55]. Furthermore, these genes were also identified to be enriched in the TGFβ signaling pathway, which has previously been associated with osteoarthritis and osteoporosis. Mutations in members of this pathway have been associated with heritable connective tissue disorders that are characterized by defects in the development and homeostasis of soft and hard connective tissue [56]. The TGFβ signaling pathways are key drivers of fibrosis, which adversely affects tendon structure, function, and chance of reinjury, and has roles in wound healing, including nonspecific scar formation and tissue-specific regeneration [57,58]. However, the role of GDFs and TGFβ in RCT requires further investigation.

Limitations of this study include a relatively small sample size. Secondly, our control samples were taken from hamstring tendon instead of age-matched supraspinatus controls that are free from disease or injury. However, the use of healthy tissue is ethically questionable, and due to the potential morbidity of taking biopsies from normal tendon, hamstring was believed to be the best compromise. Thirdly, regulation of gene expression is not solely controlled by histone modifications but rather a combination of histone modifications, histone variants, DNA methylation, remodeling enzymes, and effector proteins that influence chromatin structure, which affects a broad spectrum of other cellular processes, such as DNA replication, growth, proliferation, and repair [59]. However, this study highlights, for the first time, the potential role and effect of histone modification patterns in RCT and how this may influence the development and progression of this disorder.

## 5. Conclusions

Supraspinatus tendon samples collected from patients with RC tears had different patterns of trimethylation of H3K4 and H3K27 compared to control (hamstring) samples. Genes and pathways associated with increased or decreased trimethylation of H3K4 and H3K27 appear to have roles in regulation of skeletal muscle during myogenesis, wound healing, cell proliferation, migration, cytoskeletal organization, focal adhesion formation, tendon cell differentiation, and inflammation. Disruption to maintenance of the ECM and inflammation are often associated with musculoskeletal disorders, including tendinopathies, and it seems that RCT is not an exception. However, the role these genes and pathways play in the development, progression, and healing response in injured rotator cuffs is unclear and requires further investigation. This is the first study to characterize histone modification patterns in RCT. This work will open the door to understanding the role of histone modifications as well as the epigenome in RCT.

## Figures and Tables

**Figure 1 genes-14-00496-f001:**
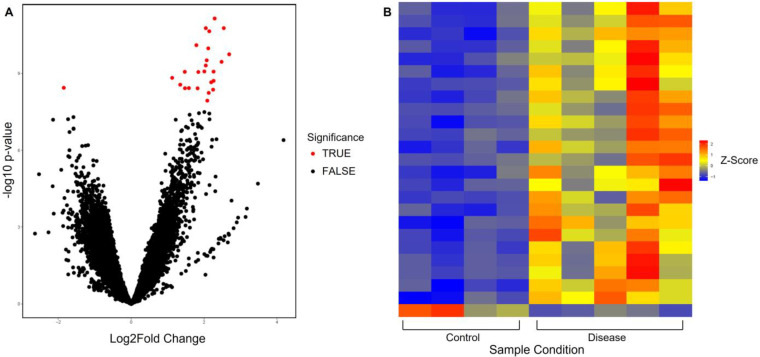
Analysis of significant peaks identified from H3K4me3 sequencing data. (**A**) A volcano plot showing significance of peaks identified for H3K4me3, where the *x*-axis is log2fold change and the *y*-axis is unadjusted *p*-value. The red dots (TRUE) indicate significant peaks (absolute log2fold > 1 and adjusted *p* < 0.05), and black dots (FALSE) indicate peaks with no significance. (**B**) Heatmap looking at the size of significant peaks identified from H3K4me3 sequencing data across sample conditions, control (hamstring), and disease (late-stage RC supraspinatus tendon).

**Figure 2 genes-14-00496-f002:**
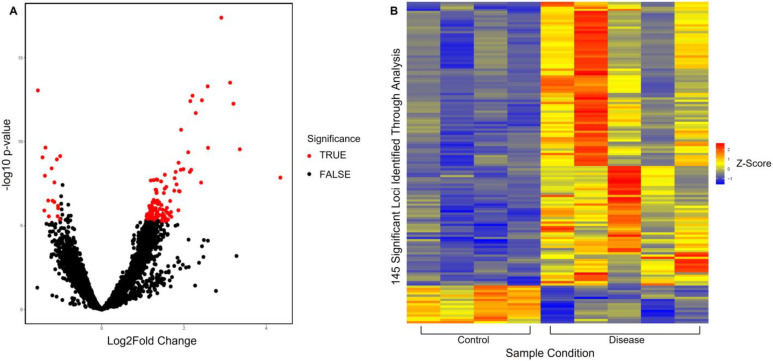
Analysis of significant peaks identified from H3K27me3 sequencing data. (**A**) A volcano plot showing significance of peaks identified for H3K27me3, where the *x*-axis is log2fold change and the *y*-axis is unadjusted *p*-value. The red dots (TRUE) indicate significant peaks (absolute log2fold > 1 and adjusted *p* < 0.05), and black dots (FALSE) indicate peaks with no significance. (**B**) Heatmap looking at the size of significant peaks identified from H3K27me3 sequencing data across sample conditions, control (hamstring), and disease (late-stage RC supraspinatus tendon).

**Figure 3 genes-14-00496-f003:**
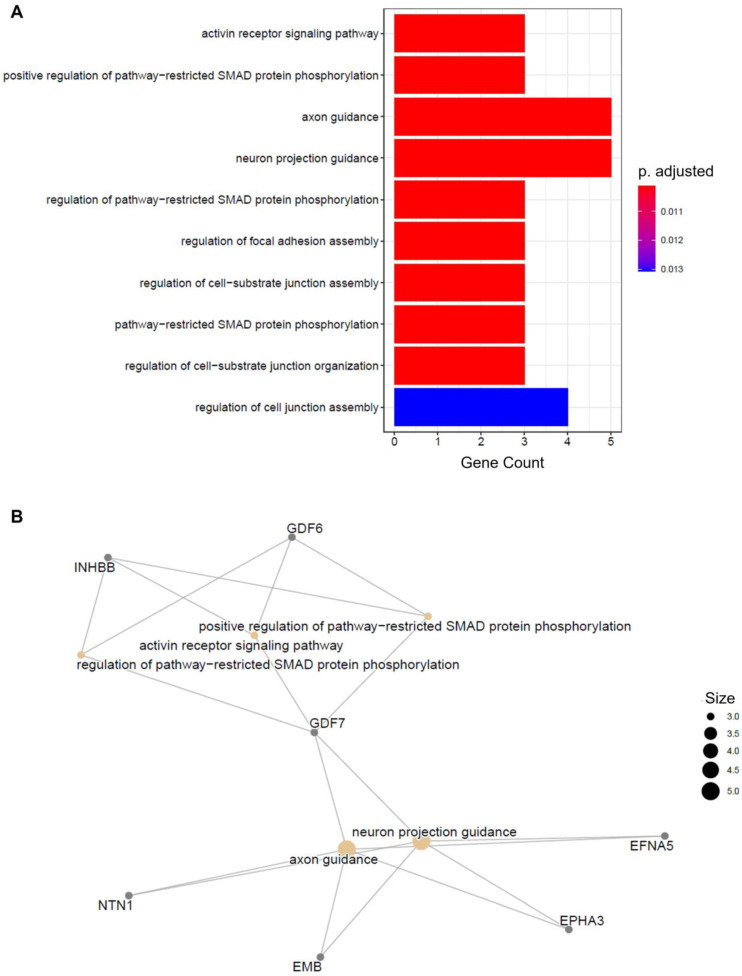
Pathway analysis for genes implicated from H3K27me3 sequencing data. (**A**) Bar plot identifying 10 pathways that were enriched in this study, with axon guidance and neuron projection guidance being the most enriched with gene counts of 5 and adjusted *p*-value of 0.011. (**B**) Gene concept network used to display the linkages of genes (gray dots) implicated in this study and biological concepts (brown dots), with size corresponding to the number of genes included within the networks.

**Table 1 genes-14-00496-t001:** DAVID functional annotation chart based on H3K27me3 gene list from peak annotation.

Category	Term	Genes	Count	%	*p*-Value	Benjamini
KEGG pathway	Axon guidance	*EPHA3*, *ROCK1*, *EFNA5*, *GDF7*, *NTN1*	5	11.6	1.0 × 10^−4^	3.5 × 10^−3^
KEGG pathway	TGF-β signaling pathway	*ROCK1*, *GDF6, GDF7*, *INHBB*	4	9.3	3.0 × 10^−4^	5.2 × 10^−3^
INTERPRO	TGF-β, N-terminal	*GDF6*, *GDF7*, *INHBB*	3	7.0	4.3 × 10^−4^	2.8 × 10^−2^
	TGF-β, conserved Site		3	7.0	1.0 × 10^−3^	2.8 × 10^−2^
	TGF-β, related		3	7.0	1.1 × 10^−3^	2.8 × 10^−2^
	TGF-β, C-terminal		3	7.0	1.3 × 10^−3^	2.8 × 10^−2^
SMART	TGFβ	*GDF6, GDF7*, *INHBB*	3	7.0	1.6 × 10^−3^	3.6 × 10^−2^
GOTERM BP	Activin receptor signaling pathway	*GDF6*, *GDF7*, *INHBB*	3	7.0	3.5 × 10^−4^	7.7 × 10^−2^
GOTERM BP	Regulation of focal adhesion assembly	*EPHA3*, *ROCK1*, *EFNA5*	3	7.0	5.6 × 10^−4^	7.7 × 10^−2^
GOTERM CC	Axon	*SRCIN1*, *CALB1*, *EMB*, *HCN1, LMTK3*	5	11.6	1.3 × 10^−3^	8.1 × 10^−2^

Table 1 A simplified table of the top 10 pathways identified through DAVID functional annotation for the H3K27me3 gene list. From these results, axon guidance was identified as the most significantly enriched pathway with a *p*-value of 0.00010 (1.0 × 10^−4^) and a Benjamini value of 0.0035 (3.5 × 10^−3^). Furthermore, the activin signaling pathway was identified as significant with *p*-value of 0.00034 (3.5 × 10^−4^) and a Benjamini value of 0.0766 (7.7 × 10^−2^), as was the focal adhesion assembly pathway with *p*-value of 0.00055 (5.6 × 10^−4^) and a Benjamini value of 0.0766 (7.7 × 10^−2^)). These pathways were also identified as significant using GOEnrich, as seen in Figure 3A and 3B. Interestingly, the TGFβ pathway was also identified as the second most enriched pathway with a *p*-value of 0.00030 (3.0 × 10^−4^) and a Benjamini value of 0.0051 (5.3 × 10^−3^).

## Data Availability

The data presented in this study are contained within the article and are available in the Appendix A section listed above.

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
