# Peer review of "Characterization of Histone Modifications in Late-Stage Rotator Cuff Tendinopathy"

_genes, 2023, doi:10.3390/genes14020496_

Round 1
Reviewer 1 Report
In this manuscript, Orchard and colleagues reported the role of histone modifications in RCT by ChIP-seqs. They found several genomic loci encoding important genes with differences in the status of H3K4me3 and H3K27me3. Importantly, they found that TGFβ signaling, axon guidance and focal adhesion formation were enriched in RCT. The results highlight the regulation of histone modifications (H3K4me3 and H3K27me3) in RCT development.
The manuscript is well-organized and well-written. The results are convincing and clear. The methods and discussion are sufficient. The conclusions are sufficiently supported by the results. I only have a few suggestions for improvements.
Minor comments:
1. For figures 1 and 2, in the volcano maps, the significant peaks (red dots) are indicated with the p. adjusted value of less than 0.05 and log2fold of more than 1. However, -log10(0.05) is around 1.3, and the cut-offs for selections in the maps are different and much higher than 1.3. Would they please indicate the cut-offs for the red dots clearly?
2. Figure 3B, what is the purpose of using the different colors for nodes (grey and brown)?
3. Are the sequencing data deposited in the public database with accession numbers?
Author Response
Please see attachment 'Response to reviewer 1'

Reviewer 2 Report
This study used chromatin immunoprecipitation sequencing for tendon samples from late-stage RC and control hamstring and carried out data analysis to identify methylation sites specific to rotator cuff tendinopathy and signaling pathways implicated in disease progression. The manuscript is clearly organized and data well organized. The conclusions are solid. There are several concerns that need to be addressed:
1) There The authors should provide justifications for choosing the two specific histones for analysis.
2) Also, the color code is not described in Figure 3B.
3) The biggest concern is the use of hamstring tendon as the control group, which significantly limits the clinical relevance of the findings.
4) As the authors mentioned, gene expression is not solely affected by histone modifications. Have the authors considered looking into the other aspects such as DNA methylation and employing other methods, given the limited amount of data presented in this paper?
Author Response
Please see attachment 'Response to reviewer 2'
